# Effectiveness and Safety of Ustekinumab for Moderate to Severely Active Crohn’s Disease: Results from an Early Access Program in Brazil

**DOI:** 10.3390/jcm11216481

**Published:** 2022-10-31

**Authors:** Julio Maria Fonseca Chebli, Rogério Serafim Parra, Cristina Flores, Antonio Carlos Moraes, Rodrigo Bremer Nones, Tarcia Nogueira Ferreira Gomes, Ana Maria Bravo Perdomo, Gustavo Scapini, Cyrla Zaltman

**Affiliations:** 1Inflammatory Bowel Diseases Center, Federal University of Juiz de Fora, Juiz de Fora 36036-900, Brazil; 2Department of Surgery and Anatomy, Ribeirão Preto Medical School, University of São Paulo (USP), Ribeirão Preto 14049-900, Brazil; 3Reference Center in Crohn and Colitis, Digestive System Institute, Rio Grande do Sul 90560-002, Brazil; 4IDOR—Institute for Research and Education, Rio de Janeio 22281-100, Brazil; 5Gastroenterology Unit, Hospital of Nossa Senhora das Graças, Curitiba 80810-040, Brazil; 6Janssen-Cilag Pharmaceutical, São Paulo 04543-011, Brazil; 7Johnson-Johnson and Internal Former Janssen-Cilag Pharmaceutical, São Paulo 04543-011, Brazil; 8Internal Medicine Department, Clementino Fraga Filho University Hospital, Federal University of Rio de Janeiro, Rio de Janeiro 21941-617, Brazil

**Keywords:** biological products, ustekinumab, crohn’s disease, early access program

## Abstract

This prospective, observational, open-label study aimed to provide access to ustekinumab prior to market authorization and assess its safety and effectiveness in patients with Crohn’s disease (CD) refractory to anti-tumor necrosis factor-α and conventional drugs in Brazil. Patients with a diagnosis of moderate-to-severe active CD for ≥3 months before screening received ustekinumab in a single intravenous induction dose (~6 mg/kg) at week 0, and a 90 mg maintenance dose, subcutaneously, every 8 or 12 weeks, from week 8 through to 80. Serious adverse events (SAE), adverse drug reactions (ADR), clinical response (per CD Activity Index and Harvey Bradshaw Index (HBI) scores), remission (per HBI scores), biomarkers (C-reactive protein (CRP) and fecal calprotectin (FC)) and endoscopic improvement rate over 80 weeks were assessed. Patients with a mean age of 39.9 years were assessed. Discontinuation rate was low (23%) and most adverse events were mild (68.7%). The SAE rate was 21% (mostly infections/infestations or gastrointestinal disorder), and ADR rate was 44%. The CD Activity Index and HBI scores decreased (by 74% and 81%, respectively) with 50% of patients showing normalized CRP and FC, and 63% achieved endoscopic improvement. Ustekinumab was fairly safe, well tolerated and effective in a Brazilian cohort of CD patients.

## 1. Introduction

Crohn’s disease (CD) is progressive chronic inflammatory disease of the gastrointestinal tract that can cause permanent damage to the intestines, leading to serious consequences. It is a disabling disease. The precise etiology of CD is not yet completely understood [1]. The prevalence of CD in Brazil has risen over time, from 0.24 per 100,000 persons in 1986 1990 to 24.1 per 100,000 persons in 2014 [2]. The CD incidence rate varies drastically across different regions of Brazil, with the Rio de Janeiro and Sao Paulo regions reporting 14.6 and 6.14 per 100,000 persons, respectively [3,4].

Current treatment strategies for CD focus on achieving long-term remission and halting disease progression. A long-term treatment approach is needed to adequately control symptoms, improve quality of life, and prevent disease-related complications [4,5]. By the time this study started, the therapeutic armamentarium for CD included conventional drugs such as salicylates (depending on guideline), antibiotics, immunomodulators, corticosteroids, tumor necrosis factor α antagonists (anti-TNF-α), e.g., infliximab, adalimumab, certolizumab pegol and infliximab biosimilar, and vedolizumab [6,7]. Although these strategies have been shown to improve the response rate and remission in patients with CD, there are a few downsides to these treatment paradigms, such as increased risk of infusion reactions, susceptibility to infections, and a risk of malignancy [8]. Furthermore, a significant number of patients fail to respond or lose responsivity. Therefore, more treatment options are necessary to enhance clinical outcomes and patient care in CD [9].

Ustekinumab (CNTO1275) is a fully human immunoglobulin G1 kappa monoclonal antibody that acts by inhibiting the p40 subunit of human cytokines interleukins 12 (IL 12) and 23 (IL 23). It prevents interaction between interleukins and the IL-12Rβ1 receptor, thereby blocking subsequent signaling and differentiation [10]. Ustekinumab has been approved by the US FDA and European Medicines Agency (EMA) for the treatment of patients with moderate to severely active CD at a dose of 6 mg/kg intravenously at baseline, followed by a 90 mg maintenance dose, subcutaneously every eight weeks [11,12].

In addition to clinical trials, ustekinumab has shown real-world efficacy and safety in treating patients with moderate-to-severe CD, supported by real-world evidence studies (ENEIDA and SUSTAIN) [13,14]. In the ENEIDA study, a nationwide Spanish registry database study included 407 CD patients treated with the recommended doses of ustekinumab, and clinical remission was achieved in 57% and 64% at 26 and 52 weeks, respectively [13], while in the SUSTAIN study [14] (331 German CD patients), clinical remission rates of 46% and 59% at 8 and 16 weeks, respectively, were observed. The SUCCESS study retrospectively evaluated 1,113 CD patients and concluded that 40% of patients achieved clinical remission, and the greatest treatment effect was seen in biologically naïve patients, with 63% and 55% of patients achieving clinical and endoscopic remission, respectively, by 12 months [15].

There is a lack of published data on the use of ustekinumab in patients with CD in Latin America. Ustekinumab was approved in Brazil in October 2009 for the treatment of plaque psoriasis and psoriatic arthritis indications [16], and in 2017 for CD treatment. As cases of CD are rising in Brazil, we designed an Early Access Program (EAP) to provide easy access to ustekinumab for patients with moderate-to-severe CD refractory or intolerant to conventional treatment and anti-TNF α therapy, and in urgent need of treatment. Open-label access to ustekinumab allowed us to collect additional safety data on ustekinumab in the Brazilian population. The primary objective of this study was to assess the safety of ustekinumab (at induction and maintenance doses) in moderate-to-severely active CD in Brazil. The secondary objective was to evaluate the effectiveness of ustekinumab by measuring clinical response and remission.

## 2. Materials and Methods

This multicenter, prospective, single arm, observational, open-label, EAP study was conducted at six sites across the South and Southeast regions of Brazil, between 17 October 2017 and 12 November 2019. The study was conducted in two phases: a screening phase and a program treatment phase. The screening phase (baseline visit) was scheduled up to 5 weeks (approximately 30 days) prior to the administration of the first dose (induction dose) of ustekinumab. The program treatment phase included an induction period, which was planned at week 0 (start of treatment phase) and a maintenance period, which occurred from week 8 through week 80 (Figure 1). During the induction period, all patients were administered a single intravenous (IV) dose of ustekinumab as per body weight (~6 mg/kg) (Appendix A). After eight weeks of induction dose, patients entered the maintenance period, wherein they received the first maintenance dose of 90 mg ustekinumab, subcutaneously. Patients with an adequate response (based on the physician’s judgement) subsequently received 90 mg ustekinumab every 12 weeks, according to label recommendations, while those with a partial response or treatment discontinuation received 90 mg ustekinumab every eight weeks through 80 weeks. The patients were followed up for 80 weeks after administration of the induction dose The safety follow-up visits occurred at the same time as the scheduled maintenance period visits: weeks 8, 16, 24, 32, 40, 48, 56, 64, 72, and 80 or weeks 8, 20, 32, 44, 56, 68 and 80, for patients in 8 or 12 weeks maintenance administration interval, respectively. A patient could be withdrawn from the program due to any of the following reasons: lost to follow up; withdrawal of consent; death; sponsor decision; an adverse event (AE) that the treating physician deemed incompatible with the continuation of treatment or lack of clinical response to treatment; or discontinuation of program treatment for any reason.

### 2.1. Patients

The study included patients (either gender) ≥ 18 years of age (at the time of informed consent) who had a confirmed diagnosis of moderate to severely active CD at least three months prior to screening. Active CD was defined as a baseline CD activity index (CDAI) score of ≥220 with one of the following: increased C-reactive protein (CRP) levels, fecal calprotectin (FC) level ≥ 250 mg/kg, endoscopic findings showing ulceration in ileum and/or colon, or Harvey Bradshaw Index (HBI) ≥ 5. Additionally, all eligible patients either showed no response or were intolerant or contraindicated to conventional treatments (e.g., immunomodulators or corticosteroids) and anti-TNF. Stable doses of concomitant CD-specific medical therapies (immunomodulators, antibiotics, corticosteroids and 5-aminosalicylic acid) were permitted. Exclusion criteria were as follows: complications of CD requiring surgery (e.g., bowel occlusion); short bowel syndrome; any type of intestinal resection within the last six months or any other intra-abdominal surgery within three months prior to the screening period; current active infection; having a draining stoma or ostomy; having or being likely to have an undrained abscess; having used an investigational drug within 30 days before the planned first dose of the program, or participated in a clinical study or treatment program of ustekinumab in Brazil.

### 2.2. Assessments

The primary endpoint was to assess the safety profile of ustekinumab. The following safety data were evaluated: AEs (including serious adverse events (SAEs), serious and non-serious adverse drug reactions (ADRs), death and other significant AEs); clinical laboratory tests (hematology, chemistry, urinalysis, and serology); and other safety observations, such as vital sign measurements, physical examinations, electrocardiogram (ECG), concomitant medication review, infusion reactions, injection site reactions, allergic reactions, and presence of tuberculosis infection. The casual relationship between an ADR and ustekinumab was assessed and classified either as “possible”, “probable” or “very likely”, per the physician’s judgement. Patients underwent colonoscopy once during the follow up period (up to 80 weeks).

The secondary endpoint was to assess the effectiveness of ustekinumab. The following parameters were measured: proportion of patients with clinical response (defined as the reduction from baseline in CDAI score of ≥100 points); proportion of patients with clinical remission (defined as HBI score < 5 or CDAI score < 150); proportion of patients with normalization of CRP and FC at weeks 16 or 20, 40 or 44, and 80.

### 2.3. Statistical Analysis

Formal sample size calculations were not performed for this descriptive study. It was expected that approximately 50 patients would be enrolled.

Statistical hypothesis tests were not performed, as this was an observational study. All safety measures were presented using descriptive statistics for the safety population, which included all patients who signed the informed consent form and received at least one dose of ustekinumab. Similarly, the effectiveness population included all patients who entered the maintenance period visit and received at least one dose of ustekinumab subcutaneously. No formal efficacy analysis was conducted. Both safety and effectiveness data were summarized as absolute frequencies and percentages (%) for categorical variables, and by measures of central tendency and dispersion for continuous variables. Additionally, percentages were calculated over the number of patients with available (non-missing) data. A sponsor’s safety team was established to review safety findings on a routine basis and to assess any new or unexpected signals. New safety findings were communicated to all sites, in accordance with local regulations and sponsor procedures.

Baseline data for almost all the measures were collected at the screening visit. For vital signs, baseline data were gathered at the induction period visit, and for laboratory evaluations, baseline data were the last values before the first study treatment.

### 2.4. Ethical Considerations

All procedures were carried out in accordance with the ethical standards of the responsible committee on human experimentation (institutional and national) and the Declaration of Helsinki 1964, as revised in 2013. This study was conducted in compliance with the Good Clinical Practice guidelines. All patients provided a signed informed consent in accordance with local, regulatory, and legal requirements prior to the initiation of any study-related procedures. According to Brazilian legislation, the Local Health Authority (ANVISA) is the authority responsible for the approval of early access treatment protocols in Brazil. The study protocol and amendment were reviewed by the Local Health Authority-Agência Nacional de Vigilância Sanitaria (ANVISA).

## 3. Results

### 3.1. Baseline Demographics and Disease Characteristics

Of the 47 screened patients, 38 met the eligibility criteria and were administered the induction dose of ustekinumab. In addition, six patients who did not meet all the eligibility requirements entered the induction period (Figure 1) presenting justified clinical necessities in agreement with medical opinions. In total, 44 patients were included in the safety analysis, and 38 patients were assessed for treatment effectiveness. The safety population comprised almost equal proportions of males (47.7%) and females (52.3%), with a mean (standard deviation (SD)) age of 39.93 (13.81) years, and with a mean BMI of 23.71 (4.90) (Table 1). At baseline, about 61.4% of patients with CD had undergone previous surgical and medical procedures, 34.1% had musculoskeletal and connective tissue disorders (Appendix A), and 70.75% of patients used concomitant immunomodulating agents or medications for the nervous system or alimentary tract and metabolism.

In total, 34 (77.3%) patients completed the EAP (weeks 0 to 80) and 10 (22.7%) patients had early discontinuation (Figure 2). Patients received a mean (SD) dose of 387.05 (71.42) mg of ustekinumab for approximately 75 min in the induction period. In the maintenance period, all patients were exposed to 90 mg ustekinumab, and the majority (65.9% patients) were allocated to an eight-week treatment regimen.

### 3.2. Safety Outcomes

During the follow-up period, 41 patients (93.2%; 95% confidence interval (CI) [85.7–100.0%]) experienced at least one AE, and a total of 249 AEs were reported (Table 2). Most events were not related to the study treatment (*n* = 30, possibly, *n* = 7, probably and *n* = 6, very likely related to treatment). Most of the AEs were mild (171 (68.7%)), and a few were moderate (69 (27.7%)), and (9 (3.6%)) patients had severe AEs. According to the system organ class (SOC) classification, gastrointestinal disorders (75.0%) were the most frequently occurring AEs, followed by infections and infestations (54.5%) and musculoskeletal and connective tissue disorders (31.8%). Per the preferred term (PT), the most frequently reported AEs (≥ 20% patients) were abdominal pain (36.4%), CD exacerbation (22.7%), and diarrhea (20.5%). Of the 249 AEs, 114 (45.6%) AEs resolved spontaneously (no action), and 108 (43.4%) AEs required pharmacological treatment. About 5.6% of AEs were deemed “serious” by the investigator and needed hospitalization or prolongation of existing hospitalization. A few AEs led to dose adjustment (6 (2.4%)), or to temporary or permanent discontinuation of treatment (3(1.2%)).

SAEs were reported by nine (20.5%; 95% CI [8.5–32.4%]) patients, and in total 14 SAEs were recorded. The most frequently reported SAEs were infections and infestations (*n* = 5, two events of liver abscess, and one event each of pneumonia, systemic viral infection, and urinary tract infection; three occurred during immunomodulators and/or corticosteroids use) and gastrointestinal disorders (*n* = 4, one event each of CD exacerbation, diarrhea, abdominal cramps (gastrointestinal hypermotility) and intestinal obstruction). Nineteen (44.2%; 95% CI [29.3–59.0%]) patients reported at least one ADR, and in total 43 ADRs were reported during the study. Six serious ADRs were reported by four patients (9.3%; 95% CI [0.6–18.0%]). A total of 18 (41.9%; 95% CI [27.1–56.6%]) patients reported at least one non-serious ADR, and a total of 37 ADRs were noted. The most frequent ADRs were infections and infestations (25.0%) and gastrointestinal disorders (20.5%), per SOC, as well as CD exacerbation (5 (11.4%) patients) and abdominal pain (3 (6.8%)) patients, per PT). From the 10 infections and infestations reported as non-serious ADRs, 4 occurred during the use of immunomodulators and/or corticosteroids.

The body mass index (BMI) values indicated a gradual increasing trend from the induction dose period through week 80 (induction period, 24.28 kg/m^2^; week 32, 24.73 kg/m^2^; week 40/44, 25.32 kg/m^2^; week 80, 25.88 kg/m^2^). Overall, hematology and chemistry laboratory values remained consistent at all study visits (Table 3 and Appendix A). The serology assessment showed negative results for human immunodeficiency virus antibody, hepatitis B surface antigen, hepatitis C antibody, and syphilis for all patients. Tuberculosis was not suspected in any patient throughout the study period. Only one (2.3%) patient had an abnormal electrocardiogram (ECG) with clinical significance (sinus tachycardia) (Table 4). No deaths were reported during the study.

### 3.3. Treatment Effectiveness

The persistence rate in weeks 16/20, 40/44 and 80 were 95% (42 of 44), 93% (41 of 44) and 77% (34 of 44) for ustekinumab. Three out of ten patients discontinued due to other reasons than lack of effectiveness or adverse events. The disease activity was reduced following administration of ustekinumab in patients with moderate-to-severe CD. The CDAI scores decreased substantially (73.86%) from baseline (287.67 points) to week 80 (75.26 points). A 70.23% reduction in mean scores was noted between week 16/20 (137.32 points) and week 32 (85.65 points), and a 24.52% reduction was observed between week 40/44 (99.63 points) and week 80 (75.26 points). Similarly, HBI scores showed a constant decrease throughout the visits, with an 81.11% decrease in the mean scores from baseline (11.54 points) to week 80 (2.18 points). At week 16/20, 73.7% (95% CI [56.9–86.6%]) of patients had clinical responses as assessed by CDAI scores, 74.3% (95% CI [56.7–87.5%]) of patients had clinical responses as assessed by HBI scores, and 76.3% (95% CI [59.8–88.6%]) of patients achieved clinical remission as assessed by CDAI and HBI scores. At the end of the study, 93.8% (95% CI [79.2−99.2%]) of patients had clinical responses, per CDAI scores; 100.0% (95% CI [88.8−100.0%]) of patients had clinical responses based on HBI scores, and 87.9% (95% CI [71.8−96.6%]) of patients achieved clinical remission as assessed by CDAI and HBI scores.

A total of 95.5% (42 of 44) patients underwent CRP evaluation at baseline, and 66.7% (28 of 42) patients reported abnormal (defined as > 5 mg/L) results. A total of 84.1% (37 of 44) patients underwent fecal calprotectin evaluation at baseline, and 83.8% (31 of 37) reported abnormal (defined as ≥ 250 mg/kg) results. The mean CRP levels decreased by 46.4% from baseline (17.67 ± 20.73 mg/L) to week 80 (9.47 ± 12.51 mg/L). The mean fecal calprotectin levels decreased by 23.7% from baseline (1055.66 ± 1141.46 µg/g) to week 80 (805.25 ± 1195.55 µg/g). At week 16/20, 38.1% (16 of 42, 95% CI [23.6−54.4%]) of patients had normalized CRP and 24.2% (8 of 33, 95% CI [11.1−42.3%]) patients had normalized FC. At the end of the study, 50% (16 of 32, 95% CI [31.9−68.1%]) and 41.4% (12 of 29, 95% CI [23.5−61.1%]) of patients had normalized CRP and FC, respectively. Twenty-seven patients underwent colonoscopy at the end of the follow-up, and endoscopic improvement was achieved in 63% (17 of 27, 95% CI [42.4−80.6%]) of patients (Table 4).

## 4. Discussion

To the best of our knowledge, this was the first multicenter, prospective, open-label study in Latin America providing in-depth insight into the safety profile and effectiveness of ustekinumab in patients with moderate-to-severe CD who were refractory to anti-TNF-α agents and conventional drugs. This EAP was designed to collect data on the specific indicators of CD, including changes in BMI, endoscopic improvement rates, biomarkers (CRP and FC), alongside routine AE and SAE profiles, clinical response, and remission rates. The overall study population age was approximately 40 years, with an equal distribution of males and females. The demographics of this study population were congruent with those of Parra et al. (2019) study, a real-world retrospective study of ustekinumab in Brazil in the refractory CD population. A similar demographic trend was also reported in an epidemiology study describing the characteristics of the IBD population in Brazil (mean age range: 37−41 years, with equal gender distribution) [17]. Furthermore, our data were aligned with nationwide studies of ustekinumab in the CD population outside Brazil [13,18,19]. Most of our patients had undergone surgical and medical procedures (61.4%) and had a history of previous treatment with immunomodulating agents (70.5%) at the start of the study, which is in line with other published studies [16,18,20]. The eight-weekly maintenance dose regimen (90 mg subcutaneously) of ustekinumab was frequently used in this study (66% patients). A similar trend was observed in a real-world evaluation of ustekinumab in Brazil in patients with anti-TNF-α refractory CD [16]. This corroborates the fact that physicians are more likely to prescribe the 8-weekly regimen of ustekinumab maintenance dose rather than the 12-weekly regimen. This is because the steady-state serum levels of ustekinumab are three-fold greater when administered every 8 weeks, demonstrating the better control of disease than the 12-weekly regimen in patients previously exposed to anti-TNF agents [7].

Overall, the discontinuation rate in this study was low (22.7%), indicating that ustekinumab was well-tolerated and effective in general, and most patients continued treatment until the study end. The discontinuation rate in our study was similar to that reported in a real-world evaluation of ustekinumab in Finland (17%) [18], while it was almost 50% lower than that reported in the IM-UNITI study (40−45% discontinuation rate), which assessed the five-year safety and efficacy of subcutaneous ustekinumab maintenance therapy in CD patients [20]. Other real-world studies have also consistently demonstrated high persistence rates (55−72%) with ustekinumab treatment in the CD population [21,22]. Overall, ustekinumab demonstrated a favorable safety profile in our study, comparable to that reported in previous clinical trials and other real-world studies of ustekinumab in patients with CD and psoriasis [4,7,8,9,10,11,12,13,14,15,16,17,18,19,20,21,23]. Although most of our patients experienced at least one AE, most of the AEs were mild in severity and were deemed not related to the ustekinumab (69%). The most frequently reported AEs were abdominal pain, CD exacerbation, and diarrhea (≥20% patients). Importantly, ADR rates in our patients were low with ustekinumab treatment. This was consistent with the IM-UNITI study and published studies in psoriatic conditions, which show that hypersensitivity reactions are rare [24] and uncommon [12] with ustekinumab administration. About 4.6% of patients reportedly developed antibodies to the ustekinumab drug in the IM-UNITI trial [25]. The incidence of serious AE was low in CD patients, with very few requiring hospitalizations (5%). This was consistent with the UNITI-1 and 2 and IM-UNITI studies, in which SAEs ranged from 5 to 12% [7].

Ustekinumab certainly has been shown to demonstrate a more desirable safety profile compared to other available treatments for CD. For example, TNF-α antagonist agents are known to be associated with immunogenic reactions, which might result in treatment failure. As per the five-year CALM study, ~25% of patients discontinued treatment with TNF-α inhibitors due to side effects [26]. Additionally, CD patients treated with anti-TNF-α often experience infusion- and injection-site reactions [27]. Treatment with certolizumab pegol, a biologic agent, has been associated with high rates of drug-related AEs (33%), AEs leading to discontinuation (11%) and SAEs (10%) [28]. The use of glucocorticoids is commonly associated with resistance and dependence in CD patients [29].

In this study, we observed a slight increase in BMI throughout the follow-up period. Although weight loss and low BMI have traditionally been associated with CD, there has been a paradoxical shift in this correlation, and a higher BMI is now recognized as a potential factor that can negatively impact the course of disease and severity [30]. In this study, the duration of follow-up was too short to identify any substantial changes in BMI that could help predict disease activity. The slight changes in BMI could be due to older age, improvement of the disease or sedentary lifestyles [30]. Further, no incidence of tuberculosis was reported following ustekinumab treatment in our study. Reports show that patients treated with anti-TNF-α agents are at high risk of tuberculosis. Ustekinumab has been shown to exhibit a 14-fold lower risk of developing tuberculosis compared to anti-TNF-α therapy [31]. Tuberculosis is a very common infectious disease in Brazil (estimated prevalence, 104 per 100,000 patient-years) leading to serious public health problems, and therefore the use of anti-TNF-α would further potentiate the risk of infection in this region [31]. Considering these facts, ustekinumab would indeed be a beneficial choice in a Brazilian population with CD.

Besides its favorable safety profile, ustekinumab exhibited excellent effectiveness in our study population. We assessed both CDAI and HBI scores, which are early indicators of treatment success or failure, to measure the clinical response. The CDAI index is a measure to quantify the severity of CD, where values ≥ 220 points are indicative of moderately to severely active disease, and <150 points are indicative of non-active disease [32]. The HBI is a simplified version of CDAI, wherein a 3-point change in the score corresponds to a 100-point change in CDAI (clinical response) and an HBI score of ≤4 corresponds to a <150 CDAI score (non-active CD) [33]. The CDAI is more useful in clinical studies [34], while HBI has been promising in real-world studies to assess disease activity [19]. In our study, the clinical response and remission rates with ustekinumab treatment were high at the end of the study (94–100% and 88%, respectively). A substantial drop of 78.8% in CDAI index scores and 81.11% in HBI index scores was observed from baseline (287.67 points and 11.54 points, respectively) to week 80 (60.70 and 2.18 points, respectively), indicating a significant reduction in disease activity. Similar results were observed in a real-world study in Brazil in patients with anti-TNF-α refractory CD [16], wherein 84.1% of patients achieved clinical response at week 8, and 75% experienced clinical remission at week 16 of 20. When compared to real-world studies outside Brazil (clinical response 50–50%, and remission 20–28%), our population showed a better clinical response and remission rates [19]. The CRP [35] and FC [36] are good biomarkers for any inflammatory bowel disease, including CD, and a reduction/normalization in their levels implies better clinical response and disease control. Almost half of the patients showed normalization of CRP levels (<5 mg/dL) and FC levels (<250 mg/kg) at week 80 in this study. Another study in the Brazilian CD population showed a similar trend in the reduction in CRP and FC following ustekinumab treatment [16]. In a five-year study of ustekinumab in CD (IM-UNITI), 38% patients reported normalization of CRP (<3 mg/L) at week 252 [20]. In recent years, the use of endoscopy has been much emphasized to help examine levels of disease activity and mucosal damage, which may help prevent recurrent surgeries in CD patients. Reports show that ~30–60% of patients attain endoscopic remission at 5–8 months after starting ustekinumab treatment [21,37]. In this study, an endoscopic improvement rate of 63% was noted after 18 months of follow-up. However, the endoscopic evaluation was solely based on the judgement of the physician caring for the patients in our study, and was not driven by the scores, as done in other published studies [36,37].

The strength of this study is that it covers a wide range of sites across the South and Southeast regions of Brazil. Additionally, we evaluated a variety of data, including routine safety and efficacy parameters and specific factors such as BMI, endoscopic evaluations, and biomarkers (CRP and fecal calprotectin), which provided a detailed understanding of the safety and effectiveness of ustekinumab in this study population. The key limitation of the study was the limited population of patients. Additionally, levels of anti-drug antibodies of ustekinumab were not assessed in our study.

## 5. Conclusions

In conclusion, ustekinumab was found to be fairly safe, tolerable and effective in patients with moderately to severely active CD refractory to anti-TNF-α agents and conventional drugs. The AEs reported were mostly mild or moderate and not related to the study drug. Almost all patients achieved clinical response and remission, and the majority showed endoscopic improvement and the normalization of CRP and FC, indicating the effectiveness of ustekinumab. More region-specific studies are needed to further confirm the benefits ustekinumab in Latin America, which can aid informed clinical decision-making and improve management strategies and patient care in CD.

## Figures and Tables

**Figure 1 jcm-11-06481-f001:**
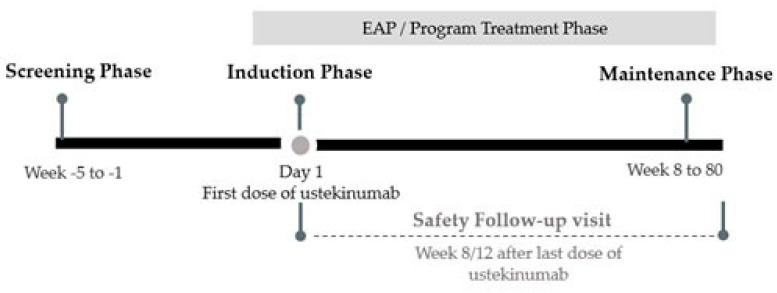
Schematic overview of program. EAP: early-access-program.

**Figure 2 jcm-11-06481-f002:**
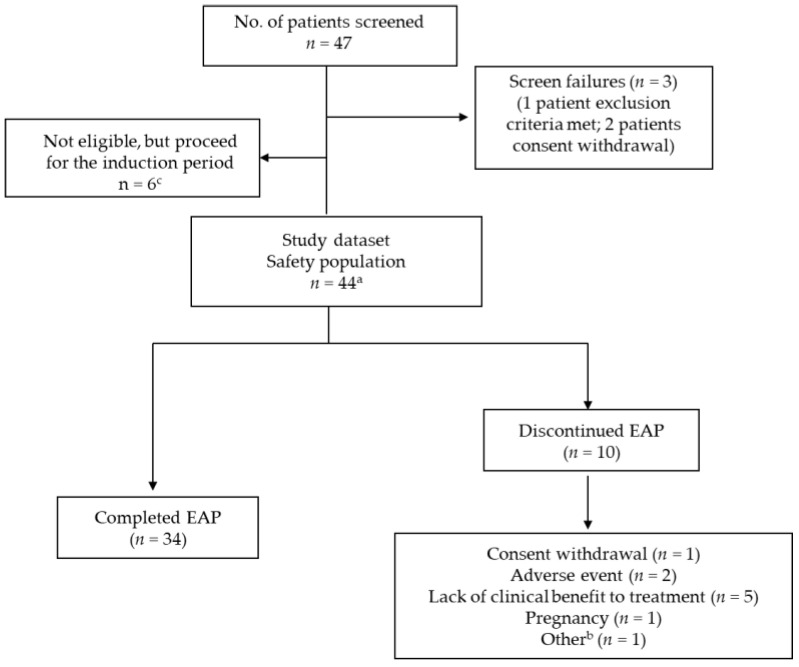
Flowchart. a: Patients who were administered at least one program treatment (CNTO1275-ustekinumab). EAP: Early access program. b: The other reason for discontinuation reported was the patient received a live vaccine and must be discontinued from study. c: Protocol deviation properly reported (patients with clinical necessities in agreement with medical and sponsor).

**Table 1 jcm-11-06481-t001:** Baseline demographic and clinical characteristics of patients with Crohn´s disease.

	Total (*n* = 44)
**Age (years)**	
N (%)	44 (100)
Mean (standard deviation)	39.93 (13.81)
**Gender, *n* (%)**	
N (%)	44 (100)
Female	23 (52.3)
Male	21 (47.7)
**Body mass index (kg/m^2^)**	
Mean (standard deviation)	23.71 (4.90)
**HBI score**	
N (%)	41 (93.2)
Mean (standard deviation)	11.54 (4.43)
**CDAI score (points)**	
N (%)	44 (100)
Mean (standard deviation)	287.67 (67.99)
**C-reactive protein levels (mg/L)**	
N	42 (95.5)
Mean (standard deviation)	17.67 (20.73)
**Previous intestinal resections**	
N (%)	8 (18.2)

**Abbreviations**: HBI, Harvey–Bradshaw Index; CDAI, Crohn’s disease activity index.

**Table 2 jcm-11-06481-t002:** Overall adverse event description.

	Participants (*n* = 44) *n* (%)	*n*AEs
**Adverse event, *n* (%)**	41 (93.2)	249
95% CI (%)	85.7, 100.0	
**Relationship with program, *n* (%)**		
Not related	-	176 (71.0)
Doubtful	-	29 (11.7)
Possible	-	30 (12.1)
Probable	-	7 (2.8)
Very Likely	-	6 (2.4)
**Severity, *n* (%)**		
Mild		171 (68.7)
Moderate		69 (27.7)
Severe		9 (3.6)
**Action taken with program patient, *n* (%)**		
None	-	114 (45.8)
Pharmacological treatment	-	108 (43.4)
Non-pharmacological treatment	-	10 (4.0)
Hospitalization or prolongation of existing hospitalization	-	13 (5.2)
Other	-	4 (1.6)
**Action taken with program drug, *n* (%)**		
None	-	232 (93.2)
Dose adjustment	-	6 (2.4)
Temporary treatment discontinuation	-	7 (2.8)
Permanent treatment discontinuation	-	3 (1.2)
Other	-	1 (0.4)
**Serious, *n* (%)**		
Yes	-	14 (5.6)
Required inpatient hospitalization or prolongation of existing hospitalization	-	14 (100)
No	-	235 (94.4)
**Ongoing, *n* (%)**		
Yes	-	45 (18.1)
Recovering/Resolving	-	13 (28.9)
Not recovered/Not resolved	-	31 (68.9)
Unknown	-	1 (2.2)
No	-	204 (81.9)
Recovered/Resolved	-	203 (100.0)
**SAE, *n* (%)**	**9 (20.5)**	**14**
95% CI (%)	8.5, 32.4	
Gastrointestinal disorders	4 (9.1)	4
Infections and infestations	4 (9.1)	5
Surgical and medical procedures	2 (4.5)	2
Cardiac disorders	1 (2.3)	1
Injury, poisoning, and procedural complications	1 (2.3)	1
Renal and urinary disorders	1 (2.3)	1
Non-SAE, n (%)	41 (93.2)	235
**ADR, *n* (%)**	**19 (44.2)**	**43**
95% CI (%)	29.3, 59.0	
Serious ADR, n (%)	4 (9.3)	6
Infections and infestations	4 (9.1)	5
Renal and urinary disorders	1 (2.3)	1
Non-serious ADR, n (%)	18 (41.9)	37
**SAE or non-serious ADR, *n* (%)**	**23 (53.5)**	**51**
95% CI (%)	38.6, 68.4	
Gastrointestinal disorders	11 (25.0)	17
Infections and infestations	11 (25.0)	15
Cardiac disorders	2 (4.5)	2
Investigations	2 (4.5)	2
Musculoskeletal and connective tissue disorders	2 (4.5)	2
Reproductive system and breast disorders	2 (4.5)	2
Surgical and medical procedures	2 (4.5)	2
Hepatobiliary disorders	1 (2.3)	1
Immune system disorders	1 (2.3)	1
Injury, poisoning and procedural complications	1 (2.3)	1
Nervous system disorders	1 (2.3)	2
Psychiatric disorders	1 (2.3)	1
Renal and urinary disorders	1 (2.3)	1
Skin and subcutaneous tissue disorders	1 (2.3)	2

**Abbreviations**: ADR, adverse drug reaction; SAE: serious adverse event; *n*AEs: number of adverse events. Serious ADRs that occurred before administration of ustekinumab were not considered for analysis.

**Table 3 jcm-11-06481-t003:** Laboratorial measurements and clinical evaluations.

	At Baseline (*n* = 44)	Week 8 (*n* = 44)	Week 16/20 (*n* = 42)	Week 40/44 (*n* = 41)	At Week 80 (*n* = 34)
**Hematology**					
**Red blood cell count performed, *n* (%)**	44 (100.0)	44 (100.0)	42 (100.0)	40 (97.6)	33 (97.1)
Mean (SD), (10^6^/mm^3^)	4.43 (0.52)	4.46 (0.55)	4.51 (0.52)	4.60 (0.54)	4.74 (0.52)
Median (Q1–Q3), (10^6^/mm^3^)	4.39 (4.02−4.69)	4.39 (4.07−4.88)	4.49 (4.24−4.78)	4.54 (4.26−5.05)	4.77 (4.45−5.00)
Normal, *n* (%)	28 (63.6)	31 (70.5)	31 (73.8)	29 (72.5)	26 (78.8)
Abnormal (clinically significant), *n* (%)	3 (6.8)	1 (2.3)	2 (4.8)	1 (2.5)	0 (0.0)
**White blood cell performed, *n* (%)**	44 (100.0)	44 (100.0)	42 (100.0)	40 (97.6)	33 (97.1)
Mean (SD), (10^3^/mm^3^)	8.71 (2.96)	8.25 (2.63)	8.37 (3.09)	7.78 (2.81)	7.59 (2.58)
Median (Q1–Q3), (10^3^/mm^3^)	8.30 (7.05−9.99)	7.17 (6.36−10.35)	7.61 (6.10−9.60)	7.10 (5.83−9.26)	6.85 (6.00−9.28)
Normal, *n* (%)	34 (77.3)	38 (86.4)	36 (85.7)	33 (82.5)	26 (78.8)
Abnormal (clinically significant), *n* (%)	1 (2.3)	0 (0.0)	0 (0.0)	0 (0.0)	0 (0.0)
**Platelet count performed, *n* (%)**	43 (97.7)	44 (100.0)	42 (100.0)	40 (97.6)	33 (97.1)
Mean (SD), (10^3^/mm^3^)	311 (123)	294 (110)	295 (107)	271 (915)	272 (871)
Median (Q1–Q3), (10^3^/mm^3^)	310 (219−383)	273 (225−344)	274 (230−344)	266 (214−326)	277 (200−336)
Normal, *n* (%)	33 (76.7)	37 (84.1)	33 (78.6)	34 (85.0)	29 (87.9)
Abnormal (clinically significant), *n* (%)	1 (2.3)	2 (4.5)	0 (0.0)	0 (0.0)	0 (0.0)
**BMI, *n* (%)**	41 (93.2)	37 (84.1)	38 (90.4)	36 (87.8)	33 (97.1)
Mean (SD), (kg/m^2^)	23.71 (4.90)	24.61 (5.36)	24.74 (5.19)	25.32 (5.58)	25.88 (5.50)
Median (Q1–Q3), (kg/m^2^)	22.19 (20.59−25.82)	23.47 (20.75−27.88)	23.90 (20.90−28.84)	24.65 (21.85−29.71)	24.43 (22.54−28.68)
Categories, *n* (%)					
Underweight	5 (12.2)	4 (10.8)	3 (7.9)	3 (8.3)	1 (3.0)
Normal weight	24 (58.5)	18 (48.6)	18 (47.4)	17 (47.2)	17 (51.5)
Pre-obesity	8 (19.5)	8 (21.6)	9 (23.7)	7 (19.4)	8 (24.2)
Obese	4 (9.8)	7 (18.9)	8 (21.1)	9 (25.0)	7 (21.2)
**CRP, *n* (%)**	42 (95.5)	-	42 (100.0)	40 (97.6)	32 (94.1)
Mean (SD) (mg/L)	17.67 (20.73)	-	11.38 (13.40)	15.78 (21.47)	9.47 (12.51)
Normal, *n* (%)	14 (33.3)	-	16 (38.1)	13 (32.5)	16 (50.0)
Abnormal ¹, *n* (%)	28 (66.7)	-	26 (61.9)	27 (67.5)	16 (50.0)
95% CI (%)			[23.6%; 54.4%]	[18.6%; 49.1%]	31.9%; 68.1%]
**Fecal calprotectin, *n* (%)**	37 (84.1)	-	33 (78.6)	32 (78.0)	29 (85.3)
Mean (SD) (mg/L)	1,055.66 (1141.46)	-	762.79 (768.08)	997.46 (1465.16)	805.25 (1195.55)
Normal, *n* (%)	6 (16.2)	-	8 (24.2)	12 (37.5)	12 (41.4)
Abnormal ², *n* (%)	31 (83.8)	-	25 (75.8)	20 (62.5)	17 (58.6)
95% CI (%)		-	[11.1%; 42.3%]	[21.1%; 56.3%]	[23.5%; 61.1%]
**CDAI, *n* (%)**	44 (100.0)	42 (95.5)	38 (90.5)	38 (92.7)	32 (94.1)
Mean (SD) (points)	287.67 (67.99)	134.03 (78.01)	137.32 (97.65)	99.63 (90.22)	75.26 (70.02)
Clinical response ^3^	-	31 (73.8)	28 (73.7)	35 (92.1)	30 (93.8)
95% CI (%)		-	[56.9%; 86.6%]	[78.6%; 98.3%]	[79.2%; 99.2%]
**HBI, *n* (%)**	41 (93.2)	44 (100.0)	38 (90.5)	41 (100.0)	33 (97.1)
Mean (SD) (points)	11.54 (4.43)	4.25 (3.21)	4.87 (4.70)	3.61 (3.79)	2.18 (2.48)
Clinical response ^4^	-	-	26 (74.3)	35 (92.1)	31 (100.0)
95% CI (%)		-	[56.7%; 87.5%]	[78.6%; 98.3%]	[88.8%; 100.0%]
Clinical remission ^5^, *n* (%)	-	44 (100.0)	38 (90.5)	41 (100.0)	33 (97.1)
Mean (SD) (points)	-	33 (75.0)	29 (76.3)	33 (80.5)	29 (87.9)
95% CI (%)		-	[59.8%; 88.6%]	[65.1%; 91.2%]	[71.8%; 96.6%]

**Abbreviations**: BMI, body mass index; CDAI, Crohn’s disease activity index; CRP, C-reactive protein; CI, confidence interval; HBI, Harvey–Bradshaw index; Q1, first quartile; Q3, third quartile; SD, standard deviation. ^1^ Abnormal CRP was defined as >5 mg/L. ^2^ Abnormal fecal calprotectin level is defined as ≥250 mg/kg. ^3^ Patients with a clinical response defined as a reduction from screening visit of ≥100 points in CDAI score. ^4^ Patients with a clinical response defined as a reduction from screening visit of ≥3 points in HBI score. ^5^ Patients in clinical remission defined as a CDAI score < 150 points or HBI < 5 points.

**Table 4 jcm-11-06481-t004:** Colonoscopy and electrocardiogram assessments.

	Result (*n* = 44)
Colonoscopy, *n* (%)	30 (68.2%)
Suggestive of inadequate control of activity, *n* (%)	10 (37.0%)
Endoscopic improvement, *n* (%)	17 (63.0%)
Electrocardiogram, *n* (%)	44 (100.0%)
Normal	30 (68.2%)
Abnormal (clinically non-significant)	13 (29.5%)
Abnormal (clinically significant)	1 (2.3%)

## Data Availability

Not applicable.

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
