# Peer review of "Effectiveness and Safety of Ustekinumab for Moderate to Severely Active Crohn’s Disease: Results from an Early Access Program in Brazil"

_jcm, 2022, doi:10.3390/jcm11216481_

Round 1

Reviewer 1 Report

This is nice real world study looking at safety and efficacy of ustekinumab in patients in Brazil from several centers who had failed anti-TNF agents for moderate to severe CD.  The multicenter nature with patient reported outcomes combined with objective measures including CRP, fecal calprotectin and endoscopy are valuable and increase the importance of the study. In the abstract, the wording can be improved. Should be 

Author Response

Dear Reviewer,

We would like to express our gratitude for the review and comments provided in the scope of the present submission.

As requested, we have provided a clean version of the document with the changes requested and another one with tracked changes and the location of the manuscript in which we addressed each suggestion from the reviewers.

RESPONSE TO REVIEWERS

Reviewer#1: This is nice real-world study looking at safety and efficacy of ustekinumab in patients in Brazil from several centers who had failed anti-TNF agents for moderate to severe CD.  The multicenter nature with patient reported outcomes combined with objective measures including CRP, fecal calprotectin and endoscopy are valuable and increase the importance of the study. In the abstract, the wording can be improved. Should be.

Response: Dear reviewer thank you for your comment and suggestion. The wording was improved.

Reviewer 2 Report

Chebli et al report the results from an early access program in Brazil concerning use of ustekinumab for moderate to severely active Crohn’s disease (CD). They claim their study to be the first multicenter, prospective, open-label study in Latin America assessing the safety profile and effectiveness of ustekinumab in patients with moderate-to-severe CD who were refractory to anti-TNF-α agents and conventional drugs.

General comment: The main limitation of the study is the small patient population, as appropriately acknowledged by the Authors in the Discussion.

Specific comments:

Abstract, line 27: please specify that 39.9 is the mean patient age, e.g., “patients with a mean age of 39.9 years were assessed”.

Patients, line 111: why 220,17? Do you mean ³220?

Patients, line 113: HBI should be ³8 for at least moderate activity. With a HBI score between 5 and 7, you also include mildly active CD patients.

- Taking a look at table S2, it appears that some patients also possibly suffered from concomitant extraintestinal manifestations (EIMs) of CD (arthralgia, skin disorders, immune system disorders, etc.). It would be interesting to know whether these EIMs responded to ustekinumab treatment as well.

- Were patients experiencing infections as adverse events taking concomitant high dose steroids and/or immunosuppressive therapy? If so, this should be specified.

Do you have data on rates of corticosteroid discontinuation at the various time points following initiation of ustekinumab treatment? How many patients achieving disease remission were corticosteroid-free?

Author Response

Dear Reviewer,

We would like to express our gratitude for the review and comments provided in the scope of the present submission.

As requested, we have provided a clean version of the document with the changes requested and another one with tracked changes and the location of the manuscript in which we addressed each suggestion from the reviewers.

RESPONSE TO REVIEWERS

Reviewer#2: Chebli et al report the results from an early access program in Brazil concerning use of ustekinumab for moderate to severely active Crohn' s disease (CO). They claim their study to be the first multicenter, prospective, open-label study in Latin America assessing the Safety profile and effectiveness of ustekinumab in patients with moderate-to-severe CO who were refractory to anti-TNF-o agents and conventional drugs.

General comment: The main limitation of the Study is the small patient population, as appropriately acknowledged by the Author s in the Discussion.

  • Abstract line 27: please specify that 39.9 is the mean patient: age.

Response: Dear reviewer thank you for your comment and suggestion. The phrase “Patients with a mean age of 39.9 years were assessed” was added to the manuscript to better understanding of the audience.

  • Patients line 111: why 220,17? Do you mean 200?

Response: It was a typo. The number was corrected as below:

  • Line 111: “Active CD was defined as a baseline CD activity index (CDAI) score of ≥220”

  • Patients line 113: HBI should be '8 for at least moderate activity. With a HBI Score between 5 and 7, you also include mildly active CD patients

Response: In this study protocol, the main inclusion criteria was CDAI which is considered the most widely used measure of clinical status in controlled clinical trials of Crohn's disease. Ustekinumab pivotal study used CDAI ≥220 as an inclusion criterion. Due to its simplicity, we understand that HBI is often used in clinical practice and real-world evidence research. Therefore, we decided to include HBI as a secondary inclusion criterion, only in patients with CDAI ≥220, together with increased C-reactive protein (CRP) levels, fecal calprotectin (FC) level ≥250 mg/kg, endoscopic findings showing ulceration in ileum and/or colon.  In the current study, all patients had CDAI over 220, being considered in moderate-to-severe disease activity. Besides, although there is correlation between CDAI and HBI, when we have CDAI slightly over 220, we can find HBI less than 8. That is why we have not included HBI above 8 as the main criteria. (Best, W.R. Inflammatory Bowel Diseases, Volume 12, Issue 4, 1 April 2006, Pages 304–310).

  • Taking a look at table S2. it appears that some patients also possibly suffered from concomitant extraintestinal manifestations (EIMs) of CD (arthralgia, skin disorders, immune system disorders. etc.). It would be interesting to know whether these ElMs responded to ustekinumab treatment as well.

Response: Thank you for your suggestion. Although we agree that having this data would be interesting and would add value to this manuscript the required data was not collect in this expanded access program.

  • Were patients experiencing infections as adverse events taking concomitant high dose steroids and/or immunosuppressive therapy'? If so, this should be specified.

Response: Thank you for your suggestion. We added the following sentences as you can see below:

  • Line 218: SAEs were reported by nine (20.5%; 95% CI: 8.5%-32.4%) patients, and in total 14 SAEs were recorded. The most frequently reported SAEs were infections and infestations (n=5, two events of liver abscess: and one event of each pneumonia, systemic viral infection, and urinary tract infection. Three occurred during immunomodulators and/or corticosteroids use) and gastrointestinal disorders (n=4, one event of each CD exacerbation, diarrhea, abdominal cramps (gastrointestinal hypermotility) and intestinal obstruction). 
  • Line 230: From the 10 infectious and infestations reported as non-serious ADRs, 4 occurred during the use of immunomodulators and/or corticosteroids.

  • Do you have data on rates of corticosteroid discontinuation in the various time points following initiation of ustekinumab treatment? How many patients achieving disease remission was corticosteroid-free?

Response: Thank you for your suggestion. We agree that having this data would be interesting and would add value to this manuscript, however, the required analysis was not the primary objective of this expanded access program. Although effectiveness was the secondary endpoint, we have not considered corticosteroids free remission in the protocol. Therefore, unfortunately, we do not have the requested result.  Considering safety analysis as the primary endpoint and the robustness of data collection regarding safety and effectiveness of ustekinumab, we believe that the absence of this analysis would not compromise the quality and importance of this manuscript.

Reviewer 3 Report

Thank you for submitting your valuable research. The authors report on a prospective study of the safety and efficacy of ustekinumab in Brazil. Although this study was well designed, collected a large number of detailed data, and is considered to be of high clinical utility, there are several concerns that needs to be solved.

1) Introduction is long and needs to be concise.

2) You should add a graph showing the change in discontinuation rate.

3)  As the primary objective of this study was to assess the safety of ustekinumab, In the DISCUSSION, the beginning should describe safety issues.

Author Response

Dear Reviewer,

We would like to express our gratitude for the review and comments provided in the scope of the present submission.

As requested, we have provided a clean version of the document with the changes requested and another one with tracked changes and the location of the manuscript in which we addressed each suggestion from the reviewers.

RESPONSE TO REVIEWERS

Reviewer#3: Thank you for submitting your valuable research. The authors report on a prospective study of the safety and efficacy of ustekinumab in Brazil. Although this study was well designed, collected, a large number of detailed data, and is considered to be of high clinical utility, there are several concerns that needs to be solved.

  • Introduction is long and needs to be concise

Response: Dear reviewer, thank you for your comment and suggestion. We agree that introduction is long, however the length of this section was necessary to include all required background information to better understanding of this study manuscript.

  • You should add a graph showing the change in the discontinuation rate.

Response: Thank you for your suggestion. We included a sentence on persistence rate at the results section as follows:

  • Line 253: “The persistence rate in weeks 16/20, 40/44 and 80 were 95% (42/44), 93% (41/44) and 77% (34/44) of ustekinumab. Three out of 10 patients discontinued due to other reasons than lack of effectiveness or adverse events.”.

  • As the primary objective of this study was to assess safety of ustekinumab, in the DISCUSSION, the beginning should describe safety issues.

Response: Thank you for your suggestion. We agree that safety is our primary endpoint. To be able to build the storytelling on the discussion, we decided to set up the context where this EAP occurred and the importance to have this kind of analysis in Brazil. Then, we addressed the discontinuation rate, which is a measure that can be correlated to both safety and effectiveness. Along the discussion, we detailed the key aspects of safety. And only in the final part, before addressing the strengths and limitations, we discussed about the effectiveness.

Round 2

Reviewer 2 Report

The paper can be accepted for publication.

Reviewer 3 Report

The authors have responded to all of the reviewer's concerns and the manuscript has been improved. Thanks indeed.